# Whole Genome Sequencing and CRISPR/Cas9 Gene Editing of Enterotoxigenic *Escherichia coli* BE311 for Fluorescence Labeling and Enterotoxin Analyses

**DOI:** 10.3390/ijms23147502

**Published:** 2022-07-06

**Authors:** Shuang Lu, Ting Tao, Yating Su, Jia Hu, Li Zhang, Guoliang Wang, Xiangyu Li, Xiaohua Guo

**Affiliations:** 1College of Life Science, South-Central Minzu University, No. 182, Minyuan Road, Hongshan District, Wuhan 430074, China; 2020044@mail.scuec.edu.cn (S.L.); 18827048202@163.com (T.T.); suyt@whammunition.com (Y.S.); m17671107321@163.com (J.H.); 2014083@mail.scuec.edu.cn (L.Z.); 2019110273@mail.scuec.edu.cn (G.W.); 2Hubei Province Nutrition Chemicals Biosynthetic Engineering Technology Research Center, Wuhan 430073, China; neal_lee@cabio.cn

**Keywords:** enterotoxigenic *Escherichia coli*, whole genome sequencing, CRISPR/Cas9, fluoresce labeling, heat-stable enterotoxin

## Abstract

Some prevention strategies, including vaccines and antibiotic alternatives, have been developed to reduce enterotoxigenic *Escherichia coli* proliferation in animal production. In this study, a wild-type strain of BE311 with a virulent heat-stable enterotoxin gene identical to *E. coli* K99 was isolated for its high potential for gene expression ability. The whole genome of *E. coli* BE311 was sequenced for gene analyses and editing. Subsequently, the fluorescent gene *mCherry* was successfully knocked into the genome of *E. coli* BE311 by CRISPR/Cas9. The *E. coli* BE311–mCherry strain was precisely quantified through the fluorescence intensity and red colony counting. The inflammatory factors in different intestinal tissues all increased significantly after an *E. coli* BE311–mCherry challenge in Sprague–Dawley rats (*p* < 0.05). The heat-stable enterotoxin gene of *E. coli* BE311 was knocked out, and an attenuated vaccine host *E. coli* BE311-ST^KO^ was constructed. Flow cytometry showed apoptotic cell numbers were lower following a challenge of IPEC-J2 cells with *E. coli* BE311-ST^KO^ than with *E. coli* BE311. Therefore, the *E. coli* BE311–mCherry and *E. coli* BE311-ST^KO^ strains that were successfully constructed based on the gene knock-in and knock-out technology could be used as ideal candidates in ETEC challenge models and for the development of attenuated vaccines.

## 1. Introduction

Enterotoxigenic *Escherichia coli* (ETEC) is one of the main causes of diarrhea in farm animals, including piglets, calves, and lambs [1], and infections by ETEC in animal husbandry and aquaculture result in severe economic losses due to high animal morbidity and mortality [2]. Generally, ETEC in an exogenous environment enters animals orally and is subsequently recognized by specific receptors in the small intestine. It attaches to the intestinal epithelium using a colonizing factor antigen (also called adhesin) and then colonizes the small intestine [3]. After this colonization, ETEC begins rapid proliferation, reaching 10^9^ colony forming units (CFU) per gram of tissue. This ETEC infection directly leads to intestinal and systemic inflammation, gut barrier dysfunction, and the subsequent growth failure of infected animals [4].

The expression of virulence factors, such as adhesin and enterotoxins, by ETEC is responsible for its pathogenicity. During the process of infection, ETEC binds to small intestinal enterocytes by adhesin and then secretes heat-labile (LT) and/or heat-stable (ST) enterotoxins, which disrupt the intestinal barrier function and cause watery diarrhea [5]. For many years, antibiotics have been widely used in animal production for the prevention and control of animal diarrhea by targeting ETEC. However, the abuse of antibiotics greatly increases safety risks by promoting antibiotic resistance. Some alternative prevention strategies, including vaccines and antibiotic alternatives [6,7,8,9], are required to reduce ETEC infection in animal production.

The ETEC strains can be modified by DNA recombinant technology to retain the desired immunogenicity while selectively eliminating toxicity, resulting in a safer and more effective vaccine suitable for commercial production [10]. The colonization factors of CS6 fimbriae have been targeted for recombinant engineered expression as potential vaccines and have provided protection against ETEC infection in the *Aotus nancymaae* diarrheal challenge model [11]. Live attenuated vaccines are also being actively pursued in vaccine research. For example, the live-attenuated ACE527 is a live, oral, multivalent vaccine that consists of three genetically attenuated and engineered strains of ETEC and contains antigens covering a wide range of ETEC surface colonization factors (CFA/I, CFA/II, and CFA/IV) as well as LT-B, the binding subunit of the LT toxin [12,13,14].

Live ETEC with toxin-free mutations can also be effective oral vaccines and can be used as recipient strains for the development of multivalent vaccines [15,16,17]. For example, the spontaneous mutant E1392/75-2A, which was engineered for the loss of the ST and LT toxins but continued to express CFA/II, provided 75% protection against a challenge with an LT^+^, ST^+^, and CFA/II^+^ ETEC strain [18]. Similarly, re-inoculation with an avirulent strain expressing adhesive fimbriae and a non-toxic form of LT provided significant short-term protection in young piglets from a challenge with a virulent ETEC strain expressing the same fimbrial adhesin and enterotoxin [19]. Therefore, ETEC-based engineered modifications are of special importance for ETEC vaccine development and enterotoxin analyses.

The evaluation of the protective efficiency of ETEC vaccine candidates requires a suitable animal infection model [11,18,19,20] that is also suitable for the development of antibiotic alternatives for in vivo anti-ETEC activities [21,22,23,24]. Reported ETEC infection models have typically used methods such as plate counting, quantitative PCR for unique virulence genes, and an enzyme-linked immunosorbent assay (ELISA) to detect and quantify ETEC [25,26]. However, ETEC is also widespread in the environment, such as in food, water, and bioaerosols [27,28,29], which renders the quantification of ETEC less specific and poorly targeted. A sensitive, selective, and rapid platform for ETEC detection is, therefore, urgently needed for challenge models.

Recently, the heterologous luciferase firefly and fluorescent protein genes have been transferred into recipient bacterial strains to provide the rapid, accurate, and visual quantification of bacteria [30,31]. Fluorescent protein genes have an advantage over reporter genes such as luciferase because the protein products spontaneously fluoresce without the addition of any substrate or cofactor [32]. The genome integration of heterologous genes has also been demonstrated to be better than plasmid-borne overexpression in terms of an increased stability and lower metabolic burden [33]. It is worth mentioning that the CRISPR/Cas9 system is one of the most powerful and revolutionary genome editing tools used to precisely manipulate the genome of various organisms [34]. Organisms of interest can now be labeled using specific-site knock-in and the integrated expression of exogenous fluorescent proteins in hosts based on CRISPR/Cas9. In the case of the ETEC infection, the underlying mechanism can be better understood by various gene modifications, including overexpression, gene deletion, or the precise regulation of virulence factors, to aid in uncovering their functions [35]. A CRISPR-Cas9 system-based continual genome editing strategy, including gene insertions and knock-outs of both single and multiple (up to three) targets in the *E. coli* genome, has been developed and is now widely applied [36,37].

In this work, our aim was to use the CRISPR-Cas9 system technology to directly modify ETEC for specific fluorescence labeling and enterotoxin analyses, based on the ability of ETEC strains to perform heterologous gene expression. The resulting modified strain was expected to serve as an attenuated host for vaccine development and as a general platform for ETEC-specific detection in ETEC challenge models.

## 2. Results

### 2.1. Identification and Characterization of E. coli BE311

The recipient strains and expression vectors that showed the confirmed expression of exogenous genes were specifically selected for further gene editing. Table 1 shows that pQE30-GFP was the most effective expression vector in the *E. coli* strains. The *E. coli* Top10, K99, and BE311 showed the strongest fluorescence as recipient strains after transformation with pQE30-GFP. The fluorescence images of colonies showed a similarly strong GFP fluorescence intensity in *E. coli* BE311 to that in *E. coli* K99 and Top10 (Figure 1A).

The 16S rRNA amplification (Figure 1B) and sequencing confirmed that the BE311 strain belonged to *Escherichia coli*. The extracted genomic DNA of BE311 was used as a template to identify the virulence factors contained in that strain. As shown in Figure 1C, the type of enterotoxins harbored in *E. coli* BE311 was tested, and the strains containing STa (219 bp), STb (155 bp), LTa (504 bp), and LTb (451 bp) were used as positive controls. Only the STa enterotoxin gene was present in BE311 (Figure 1C). The K99 (480 bp), K88 (852 bp), 987P (989 bp), and F41 (954 bp) adhesins were successively amplified and served as positive controls. The same primers were used to amplify the BE311 strain adhesins, and the results showed that *E. coli* BE311 contained a fanC adhesin identical to the one from *E. coli* K99. This confirmed the successful isolation of a wild-type ETEC, designated the BE311 strain, that harbored K99 (F5) fimbriae. The BE311 strain was deposited in the China Center for Type Culture Collection (Wuhan, China) and named *E. coli* ETEC BE311 (CCTCC NO: M2019733).

### 2.2. Whole Genome Sequencing of E. coli BE311

The whole genome sequencing showed that the chromosome of *E. coli* BE311 was 4,840,702 bp (50.74% G + C content) long and encoded 4622 genes (Figure 2A) annotated to Refseq. The whole genome sequence of the *E. coli* BE311 chromosome was deposited in GenBank with the Accession Number SUB11437696. The complete sequence of *E. coli* BE311 was compared to the VFDB database to identify potential virulence factors. The *E. coli* BE311 chromosome encodes a variety of potential virulence factors (Appendix A), including adhesions, toxins, and invasion proteins. Overall, 1364 candidate pseudogenes were identified in the *E. coli* BE311 genome, and the total length of the predicted pseudogenes was 323,478 bp (Appendix A).

Bias in the functional content distribution across the replicons in the genome was analyzed using Clusters of Orthologous Groups (COG) of proteins. The analysis identified a total of 2706 annotated putative sequences to COG functional categories and was used to predict their role in metabolism in the *E. coli* BE311 strain. Figure 2B shows that the annotation sequences of *E. coli* BE311 were subdivided into 22 classification items. In addition to some general function sequences and unknown function sequences, the top three protein functional classification items were carbohydrate transport and metabolism (287, 10.6%), amino acid transport and metabolism (274, 10.13%), and energy production and conversion (253, 9.35%). A total of 15,480 gene annotation sequences from *E. coli* BE311 were classified into three classifications of the biological process, cellular component, and molecular function, with the number and proportion of annotation sequences being 5054 (32.65%), 3863 (24.95%), and 6563 (42.40%), respectively (Figure 2C). Most of the gene sequences were involved in the cellular component classification. In addition, *E. coli* BE311 was predicted to contain 14 gene islands (Appendix A) and two CRISPR sequences (Appendix A). The Kyoto Encyclopedia of Genes and Genomes (KEGG) analysis annotated a total of 3175 genes to different pathways. The genes were mainly enriched in carbohydrate metabolism, global and overview maps, and amino acid metabolism (Figure 2D).

### 2.3. Expression and Analysis of mCherry in E. coli BE311

A labeled, stably expressed and detectable ETEC strain for use in the ETEC challenge model was obtained by the knock-in of the *mCherry* gene into the *E. coli* BE311 genome by CRISPR/Cas9, based on the information obtained by whole genome sequencing. Figure 3A shows a graphical representation of the process. The *mCherry* gene was amplified from the genome of the edited strains by a pair of primers for the original *yheO*. As shown in Figure 3B, the amplified sequence size of the positive edited strain with the *mCherry* gene knock-in was 1650 bp, while the size of the wild-type *E. coli* BE311 was 814 bp. The successful knock-in of the *mCherry* gene into the *E. coli* BE311 was verified by agarose gel electrophoresis.

The edited *E. coli* BE311 strain had no resistance gene and could not be recovered specifically. The expression of the *mCherry* gene also had to be induced by lactose or IPTG for sufficient red brightness to be visibly distinguished. The pQE30 vector was transfected into the edited *E. coli* BE311 for antibiotic resistance with the aim of ETEC recovery and quantification in the challenge model. Interestingly, the edited *E. coli* BE311 with pQE30 not only showed ampicillin resistance, but also showed red fluorescence without lactose induction, and the fluorescence was as strong as the induced fluorescence (Figure 3C). The stability of the *E. coli* BE311–mCherry containing the pQE30 plasmid was confirmed, and the strain was passaged 10 times with or without ampicillin. The results indicated that the fluorescence intensity of *E. coli* BE311 was still strong even after several generations of culture and even when cultured without ampicillin (Figure 3D). Figure 3E shows that the concentration of BE311–mCherry was highly correlated with the fluorescence intensity (R^2^ = 0.9999), indicating a relationship between the fluorescence intensity and concentration of *E. coli* BE311–mCherry–pQE30. The *E. coli* BE311–mCherry transformed with pQE30 showed obvious red fluorescence and a short bar shape under the fluorescence microscope (Figure 3F).

The *E. coli* BE311–mCherry containing pQE30 (named BE311–mCherry–pQE30) was used to challenge Sprague–Dawley rats, and the bacteria in different intestinal segments were recovered and spread on ampicillin agar plates. The resulting BE311–mCherry–pQE30 colonies showed a red brightness that could be clearly distinguished under green light excitation (Figure 3G). The ampicillin-resistant bacteria were detected in both the small and large intestines (Figure 3H,I), and the *E. coli* BE311–mCherry–pQE30 strain was readily identifiable as red colonies (Figure 3G). The examination of different intestinal segments revealed significantly increased levels of inflammatory factors (including IL-6, IL-8, and TNF-α) in the duodenum, jejunum, ileum, cecum, and colon (*p* < 0.05) after the *E. coli* BE311–mCherry challenge (Figure 3J–L).

### 2.4. Construction and Functional Analysis of ST^KO^-BE311

The virulence ST gene of BE311 was knocked out by CRISPR/Cas9, according to the process shown in Figure 4A. The genomes of wild-type *E. coli* BE311 and BE311-ST^KO^ were isolated and amplified for the ST gene. The agarose gel electrophoresis results revealed a DNA band between 100 and 250 bp in wild-type *E. coli* BE311, but it was absent in BE311-ST^KO^, indicating that ST had been successfully removed from the *E. coli* BE311 genome (Figure 4B). This knock-out of the ST gene did not interfere with the growth of *E. coli* BE311-ST^KO^, since the growth curve was similar to that of wild-type *E. coli* BE311 (Figure 4C). A comparison of the virulence and infection effects of *E. coli* BE311 and BE311-ST^KO^ cocultured with IPEC-J2 cells (Figure 4D) revealed that the BE311 challenge obviously increased apoptosis in IPEC-J2 cells (Figure 4E, *p* < 0.01), while BE311-ST^KO^ did not. Nevertheless, the deletion of ST in BE311 significantly decreased the total apoptosis rate in IPEC-J2 cells compared with wild-type BE311 (Figure 4E, *p* < 0.01).

## 3. Discussion

ETEC has a high incidence, rapid onset, and high mortality in the global pig farming industry. The diarrhea in piglets caused mainly by ETEC infection seriously affects piglet growth and production performance and is one of the major technology challenges in modern swine production [38]. The long-term use and abuse of antibiotics have resulted in a number of issues, including dysbiosis of the animals’ intestinal microbiota, the accumulation of drug residues, and environmental pollution by drugs [39]. As a result, an urgent need exists in the breeding industry for the development of effective vaccines or antibiotic alternatives against ETEC [10,13].

The targeted transformation of wild bacteria depends on the ability of the recipient strains to express exogenous genes, and this is closely related to the chosen expression vectors [40,41,42]. Green fluorescent protein is often used as an indicator to monitor heterologous protein expression in bacteria or transgenic plants [43,44]. In the present study, the recipient strains of *E. coli* were screened for their potential use as live vaccines or as indictors in challenge models. Therefore, strains with a high heterologous gene expression ability were preferentially selected for gene editing by CRISPR/Cas9 CRISPR/Cas9. As shown in Figure 1A and Table 2, both the recipient strains and the expression vectors contributed to the expression of GFP. Interestingly, the wild-type ETEC, including K99 and BE311, showed strong fluorescence protein expression, close to that of *E. coli* Top10, whereas the ability of *E. coli* K88 to express exogenous proteins was inferior to that of *E. coli* K99 and BE311. *E. coli* BE311 shared a homologous adhesin of STI to that found in *E. coli* K99. This result revealed that *E. coli* BE311 was a superior wild-type ETEC host strain for expressing exogenous genes. As a wild-type strain, the *E. coli* BE311 isolated and identified in the present study could be successfully employed in further gene editing.

The rapid development of microbial genomics has led to the use of whole genome sequencing technology for the sequencing of microbial genomic DNA on a large scale and identification of its key regulatory factors [45]. To our knowledge, no reference sequences have been reported in public databases for the whole genome sequences of K99 or F5-fimbriae-harboring ETEC. In the present study, whole genome sequencing was carried out for *E. coli* BE311 with the aim of determining the gene sequences and gene functions of *E. coli* BE311 and to discover novel virulence factors. The total length of the BE311 genome was 4,840,702 bp encoding 4622 genes (Figure 2A), which was similar to the *E. coli* strains identified in previous studies, with 5.2 to 5.5 Mbp genomes and 4633 to 5150 coding sequences (CDSs) [46,47,48]. We found that the virulence genes, ST, and fimbriae sequences of BE311 were identical to those of K99 (data not shown), suggesting that BE311 was an F5-like fimbria ETEC strain. The whole genome sequence identified in the present study will provide a novel reference for further studies on *E. coli* K99.

A study on enterohemorrhagic *Escherichia coli* (EHEC) O157:H7 has shown that the identified proteins were grouped into three important functional groups: metabolism, information storage and processing, cellular processes, and signaling [49]. Correspondingly, the BE311 genes were mainly annotated to cellular components and metabolism pathways (Figure 2A). These crucial enriched proteins may contribute to the pathogenesis of the bacterium; however, the specific gene functions of BE311 require further clarification based on more rigorous experiments. Furthermore, based on the whole genome sequence, the gRNA and the editing sites in *E. coli* BE311 could be determined for further gene knock-in and knock-out modifications by specific CRISPR/Cas9 technology [50].

In searching for inhibitors against ETEC as antibiotic alternatives, some infection experimental models are often established based on the ETEC infection [51,52,53]. The decreased colonization and proliferation of ETEC in the gastrointestinal tract is a key aim in the control of pathogen infection, making in vivo ETEC tracking particularly important. Conventional selective medium-based colony counting methods are now widely supplemented by PCR techniques, but this involves a complicated series of steps, including RNA extraction from tissues or cells, amplification, and the detection of specific virulence gene expression [54,55]. In early reports, GFP-tagged *E. coli* were constructed and successfully used to track the fate of ETEC distributed in the gut based on fluorescence signals [56,57]. These assays based on GFP-tagged *E. coli* provide a rapid and specific method for the determination of the ETEC load in challenge models. Red fluorescent proteins (RFPs) with longer emission wavelengths are more penetrating and sensitive than the green fluorescent proteins, especially for the testing of the direct imaging [58]. Therefore, mCherry was selected as the RFP in the present study for expression in *E. coli* BE311 (Figure 3A).

In previous studies, most of the bioluminescent reporter genes, such as *lux*, *luc*, *ruc*, and *gfp*, were expressed in the recipient hosts through plasmid transformation methods [31,59,60,61]. The reporter genes were, therefore, not integrated into the host chromosome and were easily lost, even without selection pressure [62]. In the present study, we constructed *E. coli* BE311–mCherry by knocking *mCherry* into the *E. coli* BE311 genome using CRISPR/Cas9 (Figure 3B). The red fluorescent protein could then be expressed after lactose induction (Figure 3C). We also increased the selectivity of *E. coli* BE311 in plate counting by transforming a vacant pQE30 plasmid to carry ampicillin resistance. Interestingly, the *E. coli* BE311–mCherry strain containing the vacant plasmid pQE30 (named *E. coli* BE311–mCherry–pQE30) showed a stable and efficient expression of the fluoresce signal even without lactose induction (Figure 3C,D). The pQE-30 vector contained a double lacO operator repression module downstream of the T5 promoter. The possible reason for the observed spontaneous fluorescence was that most of the lac repressors expressed by *E. coli* BE311 were combined into the lacO operator of pQE-30 so that the transcription and expression of mCherry was no longer repressed. This would also explain the fact that the fluorescence intensity of *E. coli* BE311–mCherry–pQE30 was not significantly enhanced after lactose induction (Figure 3C). However, we have not tested whether this effect can be achieved by incorporating a resistance cassette with the chromosomal DNA, together with the fluorescence gene. This would be a better option if the bacteria could both express fluorescence and harbor resistance, and the chromosomal mutation would be more rigorous and convincing as a theoretical control to the wild-type strain. The relative experiments will be completed in our future work.

The pathogenicity of *E. coli* BE311–mCherry–pQE30 was verified in the rat model, since the inflammatory factors were significantly increased in all intestinal segments after administration of the RFP-tagged ETEC. This result indicated that the insertion of *mCherry* did not affect the pathogenicity of the ETEC. Moreover, the RFP-tagged ETEC was easily recovered and counted on ampicillin agar plates. The ampicillin-resistant strains, except for *E. coli* BE311–mCherry–pQE30, could also be detected on the agar plates, and the RFP label helped to identify the administered ETEC, as those colonies were significantly different from the endogenous bacterial colonies (Figure 3H,I). Therefore, the *E. coli* BE311–mCherry–pQE30 constructed using CRISPR/Cas9 gene editing was an ideal tool strain that will be suitable for detecting the load of ETEC in challenge models.

Virulence factor-deficient hosts are helpful tools for ETEC vaccine exploration [63]. Heat-stable enterotoxins (STs) do not lose their activity when heated at 100 °C for 30 min, and their small size and 3D structure make them more heat resistant [64]. We recognized that the presence of ST in *E. coli* BE311 might affect its safety as a vaccine or vaccine vector; therefore, we constructed an ST knockout strain, again using CRISPR/Cas9 gene editing (Figure 4A). At the gene level, the ST gene knockout was verified, as shown in Figure 4B. The decreased apoptosis induced by *E. coli* BE311-ST^KO^ in IPEC-J2 cells indicated that the presence of ST could disrupt the function of intestinal epithelial cells to some extent, and the toxicity of *E. coli* BE311 was weakened by the knockout of the ST gene. The ETEC infection mechanism is very complicated [65]; however, the *E. coli* BE311-ST^KO^ strain constructed by CRISPR/Cas9 gene editing helped in the exploration of ST functions and could provide attenuated ETEC strains for vaccine development.

In summary, based on the screening of ETEC for its gene expression ability, we isolated a wild-type enterotoxigenic *E. coli* BE311, whose ST gene was the same as that of *E. coli* K99. The whole genome of *E. coli* BE311 was sequenced for gene editing based on CRISPR/Cas9 technology. Two strains, *E. coli* BE311–mCherry–pQE30 and *E. coli* BE311-ST^KO^, were successfully constructed based on a gene knock-in and knock-out, respectively. The two strains could represent ideal candidates for ETEC challenge model studies and attenuated vaccine development.

## 4. Materials and Methods

### 4.1. Bacterial Strains and Plasmids

Three standard ETEC strains, *E. coli* K88ab (CVCC C83901), *E. coli* K99 (CVCC C83922), and *E. coli* 987P (CVCC1523), were purchased from the China Institute of Veterinary Drug Control (Beijing, China). Four standard *E. coli* expression host strains, *E. coli* BL21, *E. coli* Rosetta, *E. coli* TG1, and *E. coli* Top10, and three expression vectors, pQE30, pET9a, and pET28a, were purchased from Solarbio Life Science Co., Ltd. (Beijing, China). Five ETEC strains were isolated from the feces of piglets with diarrhea by sorbitol MacConkey agar enrichment streaks, single colony morphology identification, 16S rRNA sequencing, and PCR identification of specific virulence genes, and kept in our laboratory. The fluorescent plasmids pUC-GFP and pUC-mCherry were kindly gifted by Dr. Xiaosheng Leung of the College of Life Science, South-Central Minzu University.

The *E. coli* strains were cultured in Luria–Bertani (LB) broth (1% tryptone, 0.5% yeast extract, and 1% NaCl) or on LB-agar plates. Bacteria harboring the pQE30 plasmid were cultured in LB medium with 100 μg/mL ampicillin. *E. coli* strains harboring the pET28a and pET9a plasmids were cultured in LB medium with 100 μg/mL kanamycin. The supplier of pCas9 (#62655) and gRNA (#62656) was Addgene (Watertown, MA, USA).

### 4.2. Screening of ETEC for Heterologous Gene Expression and Identification of Virulence Factors

The pUC-GFP plasmid was digested with *Nde* I/*BamH* I to obtain the *gfp* gene. The *gfp* DNA fragments (about 700 bp) were purified with a DNA gel extraction kit (Takara, Japan). The *gfp* fragments were then inserted into identical sites in three vectors (pET-28a, pET-9a, and pQE-30) and transformed into *E. coli* Top10 cells. Restriction enzyme identification, PCR, and sequence analysis were then used to obtain the recombinant 3 plasmids (pET28a-GFP, pET9a-GFP, and pQE30-GFP), which were then transformed into *E. coli* expression host strains and ETEC strains. On the second day, the recombinant and non-recombinant colonies were identified by directly observing their fluorescence under blue light excitation. The wild-type *E. coli* BE311 strain was selected for further CRISPR/Cas9 modification based on its strongest green fluorescence.

*E. coli* BE311 genomic DNA was extracted with a BacteriaGen DNA Kit (Cwbiotech, China), and the 16s rRNA sequence was amplified and analyzed by BLAST (https://blast.ncbi.nlm.nih.gov/Blast.cgi) (accessed on 6 March 2019) and agarose electrophoresis. The known virulence factors of ETEC and BE311 were amplified using the primers shown in Table 2, according to the reference sequences. The PCR products were verified by agarose gel electrophoresis.

### 4.3. Whole-Genome Sequencing (WGS)

#### 4.3.1. DNA Preparation

The bacterial genomic DNA of the screened *E. coli* BE311 was extracted using a DNA miniprep kit (Qiagen, Dusseldorf, Germany) according to the manufacturer’s protocol. The amounts of DNA in the samples were quantified according to Illumina sequencing sample requirements. The DNA samples were dissolved in 10 mM Tris buffer to give a 10 nM concentration in a minimum 10 μL volume. The DNA purity was determined by calculating A260/280 and A260/230 ratios in a Nanodrop ND-1000 spectrophotometer (Thermo Fisher Scientific, Wilmington, DE, USA). The quantified DNA was used to prepare genomic DNA libraries using the SQK-LSK109 ligation kit (ONT, Oxford, UK) and sequenced on a PromethION sequencer (Oxford Nanopore Technologies, Oxford, UK) and the Illumina platform (San Diego, CA, USA) using standard protocols.

#### 4.3.2. Analysis of DNA Sequence Data

The filtered reads from Hiseq were assembled with Assembler1.2. The genome was annotated and open reading frames (ORFs) identified using Prokka and/or PATRIC 3.6.5 with default settings. The WGS data sets were analyzed using an open-access bioinformatics webtool available at the Center for Genomic Epidemiology (http://www.genomicepidemiology.org/ (accessed on 6 March 2019) ). Serotype Finder 2.0 was used for in silico typing based on the WGS of the assembled genomes/contigs in FASTA format, with a threshold of 90% identity and a 60% total serotype gene length. Virulence genes were identified using Virulence Finder 2.0 with thresholds of 90% identity and 60% minimum length. Virulence genes were aligned to the VFDB database (http://www.mgc.ac.cn/VFs/main.htm). Clustered regularly interspaced short palindromic repeats (CRISPRs), genomics islands (GIs), and pseudogene candidates and prophages were determined by CRISPRfinder (https://crispr.i2bc.paris-saclay.fr/), IslandViewer 4 (http://www.pathogenomics.sfu.ca/islandviewer/), and Pseudofinder, respectively (All the websites in this section were accessed on 3–8 April 2021). The functions of the identified genes were divided into three categories: cell composition, molecular function, and biological process. The top 20 enriched GO slims with the most genes under each category were selected for plotting. After KEGG annotation, the genes were classified according to their associated KEGG metabolic pathways. 

### 4.4. Construction of mCherry-Labeled E. coli Based on CRISPR/Cas9

#### 4.4.1. Plasmid Construction

The CRISPR/Cas9 vector was constructed based on the *yheO* gene sequence published in Genbank (GenBank accession number NC_000913.3). Primers *yheO*-genomic-F and *yheO*-genomic-R were used to amplify the *yheO* gene from *E. coli* BE311 genomic DNA. The guide sequence *yheO*-gRNA was designed according to Cas9 target design principles. The CRISPR/Cas9 plasmids containing sgRNA encoding sequences were constructed using plasmid pTargetF as a template. The primers P1, P2, and P3 were designed to construct the recombinant plasmid pTargetF-sgRNA. The T5 promoter with the *mCherry* gene was obtained using primers *mCherry*-F and *mCherry*-R. The primers Left-HA-F, Left-HA-R, Right-HA-F, and Right-HA-R were used to amplify the *yehO*-Left-HA and *yehO*-Right-HA genes from *E. coli* BE311. The fragment J23119(*Spe* I)-sgRNA-gRNA scaffold was amplified by the primers SSG-F and SSG-R. These four fragments (Left-HA + *mCherry* + Right-HA + J23119 (*Spe* I)-sgRNA-gRNA) were fused by ligating the upstream and downstream editing templates using overlapping PCR. The editing template containing the Left-HA/*mCherry*/Right-HA fused sequences and the fragment J23119 (*Spe* I)-sgRNA-gRNA scaffold was digested with *Kpn* I and *Sph* I, and the target templates were inserted into the pUC57 vector. The *mCherry* gene was cloned from the plasmid pUC-mCherry. The donor DNA was obtained by mixing the *yehO* gene upstream and downstream editing templates with the *mCherry* gene and sgRNA, and then using the mixed fragment as a template to amplify the final donor DNA by overlapping PCR with double enzyme digestion. All these primers are listed in Table 3.

#### 4.4.2. Genome Editing

The pCas plasmid was transformed into *E. coli* BE311 competent cells, and the transformed cells were plated on LB agar containing kanamycin and cultured at 30 °C. *E. coli* BE311 colonies harboring the pCas plasmid were selected and used to prepare competent cells. For electroporation, 50 μL of the *E. coli* BE311 competent cells was mixed with 100 ng pTargetT series DNA and 400 ng donor DNA and subjected to electroporation in a 2 mm Gene Pulser cuvette (#1652082EDU, Bio-Rad, Hercules, CA, USA) at 2.5 kV. The product was suspended immediately in 1 mL ice-cold LB medium. The cells were allowed to recover at 30 °C for 1 h before plating on LB agar containing kanamycin (50 mg/L) and spectinomycin (50 mg/L), and incubating overnight at 30 °C. The transformants were identified by colony PCR and DNA sequencing.

The pTarget series was cured by inoculating the edited colony harboring both pCas and pTarget series into 2 mL LB medium containing kanamycin (50 mg/L) and isopropyl-β-D-thiogalactopyranoside (IPTG; 0.5 mM). The culture was incubated for 16 h, diluted, and plated on LB agar containing kanamycin (50 mg/L). The colonies were confirmed as cured by determining their sensitivity to spectinomycin (50 mg/L) and were then used in a second round of genome editing. The pCas series was cured by growing the colonies nonselectively overnight at 37 °C [66].

Figure 3A outlines the process used for the construction of the *E. coli* BE311 expression system with mCherry fluorescent protein (named *E. coli* BE311–mCherry) as the reporter gene. A blank vector pQE30 was transformed into the *E. coli* BE311–mCherry to test expression with or without inducement. The strains were cultured in 37 °C in LB for 12 h and then induced by addition of 5 g/L lactose and incubation at 18 °C for 10 h. The fluorescence of the different strains was observed in daylight after centrifugation.

#### 4.4.3. Verification of the *E. coli* BE311–mCherry Strain

The effects of *mCherry* labeling in *E. coli* BE311 were tested by PCR using primers 5′-GTAGAAGCCGGTAAAGGCGA-3′ and 5′-AGAAACGCTGCTATCCGCTC-3′. The genomic DNA of *E. coli* BE311 and recombinant strain *E. coli* BE311–mCherry were used as templates. The final PCR reaction volume was 25 μL, and included 0.5 mmol/L primers, 1.5 mmol/L Mg^2+^, 200 mmol/L dNTPs, 50 ng of template DNA, and appropriate ddH_2_O. The PCR reaction program began with initial denaturation at 98 °C for 3 min, followed by 35 cycles of denaturation at 98 °C for 10 s, annealing at 60 °C for 20 s, and extension at 72 °C for 7 s. The final extension was held at 72 °C for 5 min. The resulting PCR products were separated on 1% agarose gel and subjected to DNA sequencing. The PCR product size of BE311 genome was 814 bp, and the product size amplified from the mCherry-positive genome was 1650 bp.

#### 4.4.4. Fluorescence Intensity Determination of *E. coli* BE311–mCherry

The *E. coli* BE311–mCherry from the fermentation broth was harvested for analysis by centrifugation (6000× *g* for 5 min), washed twice, and suspended in physiological saline. The suspensions of *E. coli* BE311–mCherry with the initial optical density were diluted about tenfold at a gradient of 2 times and the fluorescence intensity was determined. The suspensions were plated for counting, and the linear correlation between strain concentrations (CFU/mL) and the fluorescence intensity (AU) was established.

A fresh bacterial solution of *E. coli* BE311–mCherry was washed twice with sterile double distilled water, and the excitation and emission wavelengths of mCherry at 587 nm/610 nm, respectively, were used to determine the fluorescence intensity of *E. coli* BE311–mCherry in an F-7000 fluorescence spectrophotometer (Hitachi, Ltd., Tokyo, Japan). A sample of the bacterial solution was evenly coated on a clean glass slide, and the fluorescence was observed under a fluorescence microscope (Nikon Eclipse C1, Minato, Japan).

### 4.5. Construction of an ST-Knock-Out E. coli BE311 Strain Based on CRISPR/Cas9

#### 4.5.1. Plasmid and Editing Template DNA Construction

The guide RNA and editing template were designed based on the whole genome sequence of *E. coli* BE311 obtained in this study. The ST gene sequence was used to design the corresponding gRNA in the open website (http://www.rgenome.net/cas-offinder/) (accessed on 8 June 2021); its sequence was 5′-CAGTTGACTGACTAAAAGAG-3′. The gRNA was synthesized and linked to the plasmid pgRNA. The sequence of *E. coli* BE311 was then used to insert the sequence of the left and right homologous arms of the ST gene into pUC57 to obtain the editing template. Primers 5′-ATGAAAAAGCTAATGTTGGCAATTT-3′ and 5′-TTAATAACATCCAGCACAGGCAGGA-3′ were used to amplify the homologous arm sequences from pUC57-ST. The PCR products were used as the donor DNA.

#### 4.5.2. Genome Editing with Chemical Transformation

*E. coli* BE311 competent cells containing pCas were prepared and electroporated by mixing 100 μL competent cells with 100 ng pgRNA and 400 ng donor DNA, and electroporating in a 2 mm Gene Pulser cuvette (Bio-Rad Korea, Seoul, Korea) at 2.5 kV. The product was suspended immediately in 700 μL ice-cold LB medium. The cells were allowed to recover at 30 °C for 1 h before plating onto LB agar containing kanamycin (50 mg/L) and ampicillin (50 mg/L), and incubation overnight at 30 °C. Positive clones were identified by colony PCR. The positive strains were heat treated to cure pCas and pgRNA.

#### 4.5.3. Verification of the Editing Efficiency

The genomic DNAs of wild-type and engineered *E. coli* BE311 (named BE311-ST^KO^) were prepared from 4 mL scale overnight cultures in LB medium using a bacterial genomic DNA extraction kit (Cwbiotech, Taizhou, China). The ST gene was amplified using the primers 5′-CGAACAAGAAAAGGATAAAACCA-3′ and 5′-TATAAAAGCGTTGTTTTTTATTA-3′.

### 4.6. Stability Detection of E. coli BE311–mCherry and BE311-ST^KO^

Overnight cultures of *E. coli* BE311–mCherry and *E. coli* BE311-ST^KO^ were transferred to fresh LB medium at an inoculum volume of 1% and incubated at 37 °C, 200 rpm/min. The bacterial solution was passaged, continuously, 10 times, each time at an interval of 24 h, diluted, and the OD_600 nm_ absorbance and fluorescence intensity were measured.

### 4.7. E. coli BE311–mCherry–pQE30 Challenge In Vivo

Twelve male Sprague–Dawley rats were randomly divided into two groups: a control group and the ETEC challenge group. Rats in the ETEC group were intragastrically administered with 1 mL (10^6^ CFU) *E. coli* BE311–mCherry–pQE30 daily via oral gavage for 7 days. The rats in the control group were orally gavaged with 1 mL isopycnic phosphate buffered saline (PBS) daily for 7 days. All rats were sacrificed at day 8, and the intestines were separated into five gastrointestinal segments, including the duodenum, jejunum, ileum, cecum, and colon. Each intestinal sample was completely suspended in PBS, and then homogenized, filtered through sterile gauze, and diluted 10-fold with normal saline. The numbers of anti-Amp^+^ and BE311–mCherry bacteria were calculated by colony counting on LB agar plates containing 50 mg/L ampicillin. The expression of IL-6, IL-8, and TNF-α in different intestinal segments was tested with an ELISA kit (CAMILO Biological, Naijing, China), according to the kit instructions. The animal experiment was approved by Scientific Ethics and Safety Committee of South-Central Minzu University (Ethical permission number: 2020-SCUEC-006).

### 4.8. Flow Cytometry Analysis of Apoptosis in Cells Infected with E. coli BE311-STKO

Apoptosis was assessed using an Annexin V-FITC/PI double-stain assay kit (Beyotime, Haimen, China), according to the manufacturer’s instructions. Briefly, the IPEC-J2 cells were seeded in 60 mm dishes and cultured with PBS or *E. coli* BE311 and BE311-ST^KO^ for 6 h. The cells were then collected and stained in binding buffer with 5 μL polyimide (PI) solution and 10 μL FITC-conjugated annexin V for 15 min in the dark at room temperature. Apoptotic cells were detected with a CytoFLEX flow cytometer (BD, Franklin Lakes, NJ, USA), and the data were analyzed with the FlowJo software program (Version 10, FlowJo, Ashland, OR, USA).

### 4.9. Statistical Analysis

Student’s *t* test and one-way ANOVA statistical analysis were used for statistical comparisons of differences between two or more than two groups, respectively. Column charts were constructed from the mean ± SD values using GraphPad Prism 6.0 (Graphpad Inc., San Diego, CA, USA). Values of *p* < 0.05 were deemed statistically significant. All experiments were performed as at least three independent replicates.

## Figures and Tables

**Figure 1 ijms-23-07502-f001:**
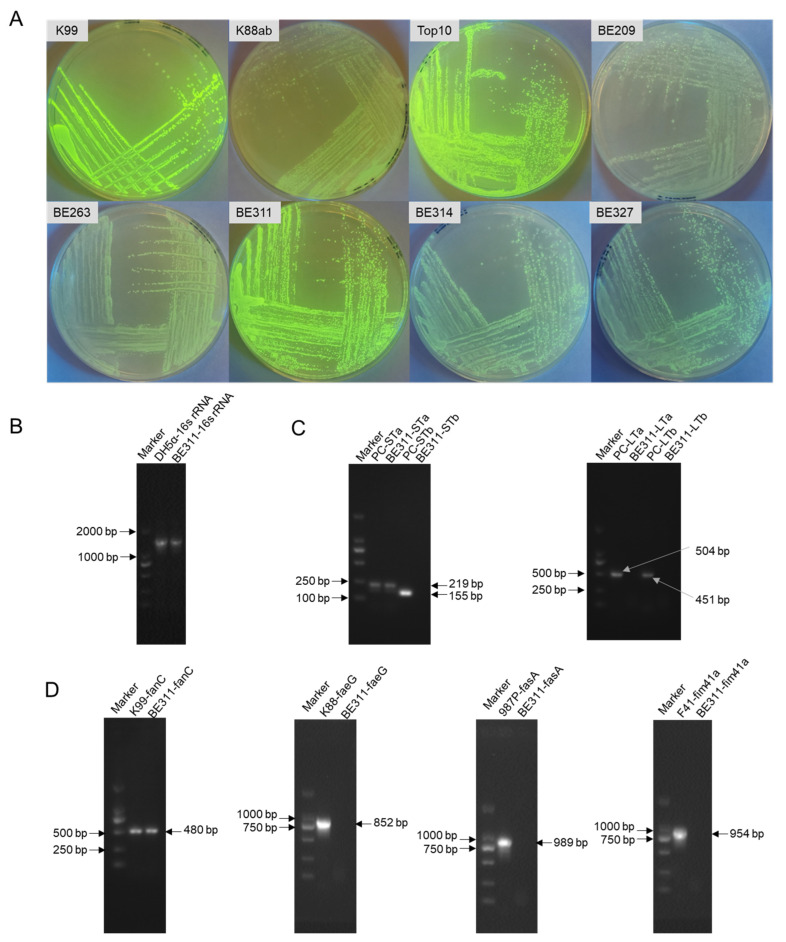
Identification and characterization of *E. coli* BE311. (**A**), The expression of pQE30-GFP in different *E. coil* strains. (**B**), PCR amplification of the 16S rRNA of *E. coli* BE311. (**C**), Identification of the enterotoxins in *E. coli* BE311 by agarose gel electrophoresis (AGE). (**D**), Identification of the adhesions in *E. coli* BE311 by AGE.

**Figure 2 ijms-23-07502-f002:**
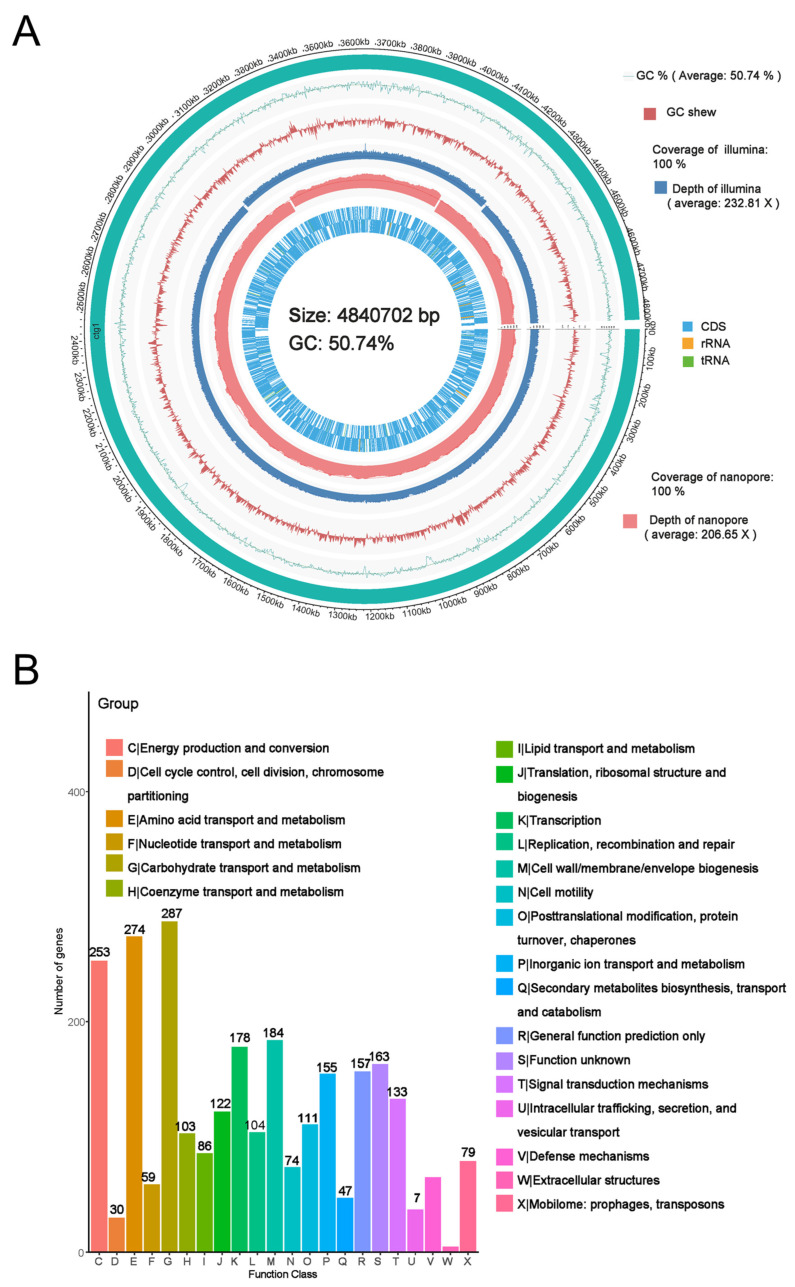
Whole genome sequencing of *E. coli* BE311. (**A**), The circus plot of *E. coli* BE311 genome. (**B**), The COG analysis of *E. coli* BE311 genome. (**C**), GO analysis of *E. coli* BE311 genome. (**D**), KEGG analysis of *E. coli* BE311 genome.

**Figure 3 ijms-23-07502-f003:**
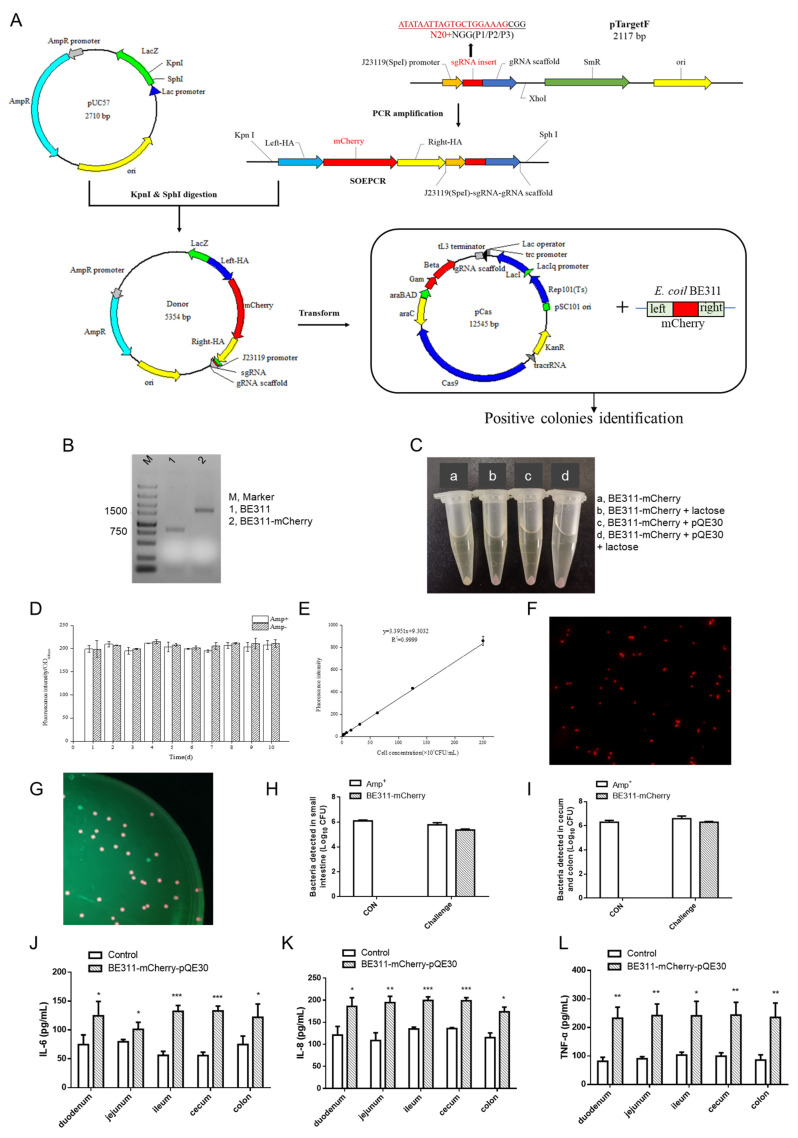
Expression and analysis of mCherry in BE311. (**A**), The process of knocking *mCherry* into BE311 based on CRISPR/Cas9. (**B**), Agarose gel electrophoresis detection of the *mCherry* in BE311 genome. (**C**), Red fluorescence expression in different strains with or without lactose inducement. (**D**), Stability analysis of BE311–mCherry–pQE30 with or without ampicillin. (**E**), Construction of the correlation curve between fluorescence intensity and concentration of BE311–mCherry–pQE30. (**F**), The fluorescence of *E. coli* BE311–mCherry–pQE30 captured under fluorescence microscope. (**G**–**I**), The gut microbiota of *E. coli* BE311–mCherry–pQE30-challenged mice was obtained and cultured in LB plate for 24 h. The fluorescence of colonies and bacteria in intestines (**H**,**I**) was identified under blue exciting light and fluorescence microscope. (**J**–**L**), The expression of IL-6, IL-8, and TNF-α in different intestines of *E. coli* BE311–mCherry–pQE30-challenged SD rats. Statistical significance was determined using Students’ *t* test. * *p* < 0.05, ** *p* < 0.01, *** *p* < 0.001.

**Figure 4 ijms-23-07502-f004:**
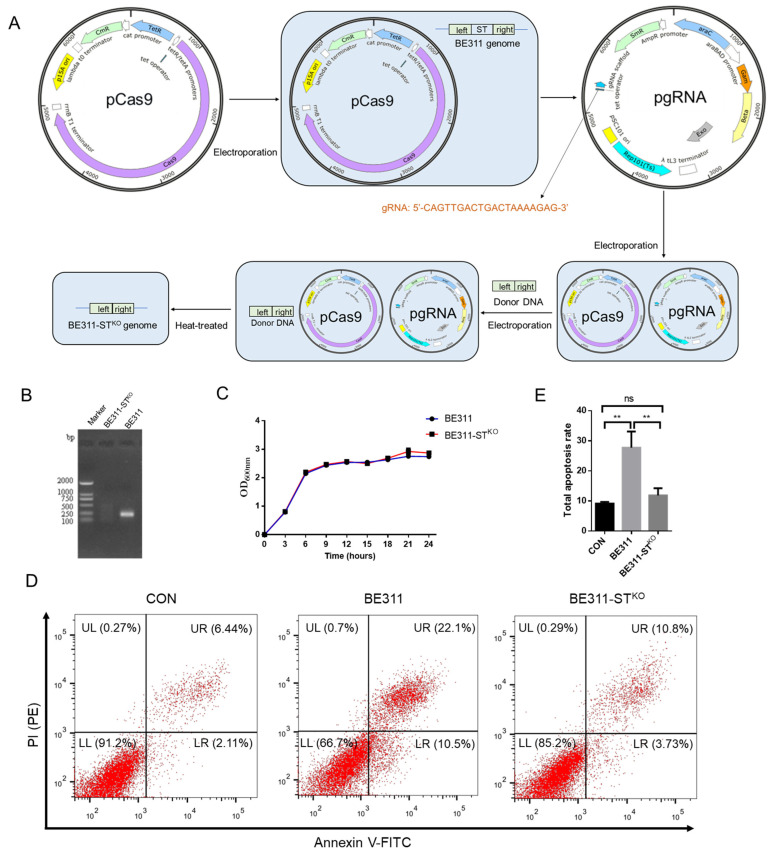
Construction and analysis of BE311-ST^KO^. (**A**), The knock-out process of ST gene in BE311. (**B**), Amplification of ST gene in wild-type and ST-knock-out BE311 strains. (**C**), The growth curve of BE311 and BE311-ST^KO^ strains within 24 h. (**D**), IPEC-J2 cells were challenged with BE311 or BE311-ST^KO^ for 6 h. Then the cellular apoptosis was examined by flow cytometry. (**E**), The total apoptosis rate in IPEC-J2 cells challenged with BE311 or BE311-ST^KO^ for 6 h. ** *p* < 0.01, ns, no significant difference.

**Table 1 ijms-23-07502-t001:** The GFP-expressing ability of different *E. coli* strains and vectors.

*E. coli*	Strains	pQE30-GFP	pET28a-GFP	pET9a-GFP
Standard *E. coli* recipient hosts	E. coli BL21	—	—	—
	*E. coli* Rosetta	++	—	—
	*E. coli* TG1	++	—	—
	*E. coli* Top10	+++	—	—
Standard ETEC	*E. coli* K88ab	+	—	—
	*E. coli* K99	+++	—	—
	*E. coli* 987P	/	/	/
Wild-type ETEC	*E. coli* BE209	+	—	—
	*E. coli* BE263	++	/	/
	*E. coli* BE311	+++	—	—
	*E. coli* BE314	++	/	/
	*E. coli* BE327	++	+	/

—, no or suspicious green fluorescence; +, low green fluorescence intensity; ++, middle green fluorescence intensity; +++, high green fluorescence intensity; /, no colonies recovered.

**Table 2 ijms-23-07502-t002:** Primers used for PCR.

Primer	Primer Sequences (5’-3’)	Product Size (bp)	GenBank Accession No.
LTa-F	ATGATTGACATCATGTTGCATATAG	451	V00275.1
LTa-R	TATGGGTGAGGGCTGTAT		
LTb-F	CGCGGATCCCCAGACTATTACAGAACTA	504	M17873.1
LTb-R	ATAAGAATGCGGCCGCAAGCTTGCCCCTCCAGCCTAGC		
STa-F	ATGAAAAAGCTAATGTTGGCAATTT	219	M25607.1
STa-R	TTAATAACATGGAGCACAGGCAGGA		
STb-F	CATCTACACAATCAAATAA	155	AY028790.1
STb-R	TTAGCATCCTTTTGCTGCAA		
K88-F	CGGGATCCATGAAAAAGACTCTGATT	852	AJ616236.1
K88-R	CCGGTACCTTAGTAATAAGTAATTGC		
K99-F	CGCGGATCCAATACAGGTACTATTAAC	480	M35282.1
K99-R	CGCAAGCTTTTACATATAAGTGACT		
987P-F	CGCGGATCCAAATTTAGAAAAGTGCAT	989	M35257.1
987P-R	CGCAAGCTTATATAAACAAAAACACAA		
F41-F	CGCGGATCCTGAATCCGCAGGGGATGG	954	M21788.1
F41-R	CGCAAGCTTCTCACTGCCCCCAACTAC		
16S rRNA-27F	AGAGTTTGATCCTGGCTCAG	1453	
16S rRNA-1492R	TACGGCTACCTTGTTACGACTT		
GFP-F	GCGGGATCCATGAGTAAAGGAGAAGAAC	717	X83959.1
GFP-R	GCGAAGCTTCTATTTGTATAGTTCATCC		

**Table 3 ijms-23-07502-t003:** Primers used in the construction of mCherry-labeled *E. coli* BE311.

Primers	5′-3′ Sequence
*yheO*-genomic-F	GTAGAAGCCGGTAAAGGCGA
*yheO*-genomic-R	AGAAACGCTGCTATCCGCTC
*yheO*-gRNA	ATATAATTAGTGCTGGAAAGCGG
P1	ATATAATTAGTGCTGGAAAGGTTTTAGAGCTAGAAATAGCA
P2	AGTCCTAGGTATAATACTAGTATATAATTAGTGCTGGAAAG
P3	CTTATGGAGCTGCACATGAACTCGAGTAGGGATAACAGGGT
Left-HA-F	GGTACC GTAGAAGCCGGTAAAGGCGAAGC
Left-HA-R	AGCAAATAAATTTTTTATGATTATATCCGGCCTGTAATAA
mCherry-F	TTATTACAGGCCGGATATAATCATAAAAAATTTATTTGCT
mCherry-R	TAATACAGCGGAGGTTCCGCCTTGTACAGCTCGTCCATGC
Right-HA-F	GCATGGACGAGCTGTACAAGGCGGAACCTCCGCTGTATTA
Right-HA-R	AGGACTGAGCTAGCTGTCAAATCTTCCGGCCTGTATGTTCACC
SSG-F	GGTGAACATACAGGCCGGAAGATTTGACAGCTAGCTCAGTCCT
SSG-R	GCATGC GCACCGACTCGGTGCCACTTTT

## Data Availability

The data presented in this study are openly available in Genebank at SUB11437696.

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
