# Peer review of "Whole Genome Sequencing and CRISPR/Cas9 Gene Editing of Enterotoxigenic Escherichia coli BE311 for Fluorescence Labeling and Enterotoxin Analyses"

_ijms, 2022, doi:10.3390/ijms23147502_

Round 1

Reviewer 1 Report

Comments to the Authors of manuscript number: ijms-1768364 entitled “Whole genome sequencing and CRISPR/Cas9 gene editing of enterotoxigenic Escherichia coli BE311 for fluorescence labeling and enterotoxin analyses”.

It seems that the study is only performed on ETEC. But, Authors used animals. Firstly, ETEC was collected from piglets, than rats were used to the part of in vivo. Authors use very good technology and present data which are worth to be published, but the ethical permission is needed to use animals, which should be presented.

Bellow are other comments:

1. what about Post-weaning diarrhea as an example of endogenous ETEC?

2. P 2 L 4 – reference is needed

3. “ETEC is widespread in the environment in vivo” what do Authors mean by the environment in vivo?

4. P2 L 39  it is worth to mention that the CRISPR/Cas9 system is one of the most powerful and revolutionary genome editing tools uses to precisely manipulate the genome of various organisms.

5. P 3 L 9-10 this sentence suits rather to the discussion

6. Authors inform in Materials and methods, that five suspicious ETEC were isolated from the feces of piglets with diarrhea. How it was performed and where is the number of ethical permission?

7. How it is known that it was exogenous infection?

8. P 7 Paragraph 3 “The ampicillin-resistant bacteria were detected in both the small and large intestines (Figure 3H, 3I)..”

Authors did not mention about the ethical permission needed to perform this part of the study presented in the paper. Of course, it is described in Materials and methods (4.7), but the number of the ethical permission is needed.

9. the part of 4.7: all these segments have to be listed: the duodenum, jejunum, ileum, cecum, and colon.

Reviewer 2 Report

The manuscript by Lu S., et al. describes methodologic techniques in gene manipulation of wild-type ETEC E. coli. The authors first created a lactose -inducible fluorescent E. coli strain by incorporating mCherry gene to the chromosomal DNA by CRIPR/Cas9. Secondly, they created a constitutively-fluorescent E. coli in which fluorescence was expressed from the transfected plasmid, pQE30-mCherry::amp. Thirdly, they generated a ST-null mutant by using CRIPR/Cas9. The paper could be referred by many researchers who seek molecular techniques to successfully generate mutants in wild-type E. coli strains of their interest.

Comments:

Table 1. In the seek for the most effective expression vector in wild-type E. coli, pQE30 was found to be the best vector among the three vectors tested. However, expression from pQE30 was detected only in two of the five wild-type strains tested. Does the result indicate that GFP-pQE30 cannot be used in a wide range of wild-type strains? It would be quite attractive for many of the readers who are willing to transform GFP in E. coli, if the vector could be used with minimum strain specificity.

Figure 2. The text in the figure is too small and too low in resolution to read, at least in my print.

Page 7, line 8. The authors explain that the chromosomally-edited BE311 strain could not be recovered specifically because it had no resistance markers, and that this was one of the reasons they expressed fluorescence from the plasmid pQE30. It could be possible to incorporate a resistance cassette to the chromosomal DNA together with the fluorescence gene. Another way to distinguish a non-selective mutant is to do so by determining an accompanied silent mutation, if any. Did the authors not try other selection approaches, and if so why was the plasmid expression system encouraged more than the chromosomal mutation? Chromosomal mutation will be a theoretical control to the wild-type strain, but not exactly the plasmid-derived mutant, although pQE30 was accidentally found to be relatively stable even without selection pressure.

Figure 3A and Figure 4A. The text in the figures is too small and too low in resolution to read, which is a disappointment for the readers who are willing to refer to the described gene-manipulation method.

Round 2

Reviewer 1 Report

I have no more comments. Thank Authors for answers.